# The Role of Semaphorins in Metabolic Disorders

**DOI:** 10.3390/ijms21165641

**Published:** 2020-08-06

**Authors:** Qiongyu Lu, Li Zhu

**Affiliations:** Cyrus Tang Hematology Center, Suzhou Key Laboratory of Thrombosis and Vascular Diseases, Soochow University, Suzhou 215123, China; luqiongyu@suda.edu.cn

**Keywords:** semaphorins, metabolic disorders, obesity, diabetic complications

## Abstract

Semaphorins are a family originally identified as axonal guidance molecules. They are also involved in tumor growth, angiogenesis, immune regulation, as well as other biological and pathological processes. Recent studies have shown that semaphorins play a role in metabolic diseases including obesity, adipose inflammation, and diabetic complications, including diabetic retinopathy, diabetic nephropathy, diabetic neuropathy, diabetic wound healing, and diabetic osteoporosis. Evidence provides mechanistic insights regarding the role of semaphorins in metabolic diseases by regulating adipogenesis, hypothalamic melanocortin circuit, immune responses, and angiogenesis. In this review, we summarize recent progress regarding the role of semaphorins in obesity, adipose inflammation, and diabetic complications.

## 1. Introduction

Semaphorins are a large family of proteins involved in different physiological and pathological processes. Semaphorins have been discovered from viruses, in insects to mammals, and are expressed in most tissues [1]. As illustrated in Figure 1, the semaphorin family includes 30 proteins that are divided into eight classes based on structural features and distribution among different phyla [2]. Class-1 semaphorins [3] and Class-2 semaphorins [4] are found only in invertebrates, while Class 3–7 semaphorins are found only in vertebrates [5] (except for Sema5C, which is also found in invertebrates [6]). Semaphorin V members express in viruses. All semaphorins contain an approximate 500 amino acid extracellular Sema domain [7]. In addition to the Sema domain, semaphorins also contain the plexin-semaphorin-integrin (PSI) domain, and distinct protein domains that are expressed by different subclasses, including immunoglobulin-like (Ig), thrombospondin, and basic C-terminal domains [8]. The receptors for semaphorins include plexins, neuropilins [9], and other molecules, such as integrins, proteoglycans, and receptor tyrosine kinase(RTKs) [10]. Plexins, which are found in vertebrates and fall into Classes A–D, also contain the extracellular Sema domains [11]. Several recent studies on the structure of semaphorins confirmed that each Sema domain of a semaphorin homodimer binds to a Sema domain of plexins to promote plexin dimerization for signal transduction [11,12,13].

Among the five semaphorin classes in vertebrates, the Class-3 semaphorin subfamily (Sema3A–3G) is the most studied semaphorins in metabolic disorders from obesity to diabetic complications. Class-3 semaphorins are the only secreted vertebrate semaphorins. In addition to the Sema domain, like other semaphorin members, all Class-3 semaphorins contain at least two conserved basic cleavage sites for furin-like pro-protein convertases (FPPC). A major cleavage site in Sema3 is located downstream of the Sema domain and the other one is located in the basic domain [14]. There are two subfamilies of receptors for Class-3 semaphorins, i.e., neuropilins and plexins. Most of the Class-3 semaphorins, except for Sema3E [15], utilize neuropilin 1 (Nrp1) or neuropilin 2 (Nrp2) or both as their main binding receptor [16,17,18,19]. Neuropilins are not sufficient to transduce Class-3 semaphorin signals due to their short intracellular domains. The neuropilins form complexes with one or more of the four type-A plexins or with PlexinD1 [9,18,20]. In these functional semaphorin receptor complexes, the plexins serve as the signal transduction components [21].

For Class-4 semaphorins, the main receptors are PlexinB molecules. They also bind PlexinC1 and PlexinD1. Neuropilins are found as co-receptors for semaphorins in some conditions. For example, the main receptors for Sema4D, the most studied Class-4 semaphorins, are PlexinB1 [22,23,24] and PlexinB2 [25,26,27]. CD72 also mediates the function of Sema4D in some immune cells [28,29]. PlexinA and PlexinB molecules mediate the most biological functions of Class-5 semaphorins. PlexinA1, -2, and -4 are the main receptors for Class-6 semaphorins [30]. For the only member of Sema7, PlexinC1 [31] and integrin β1 [32] are the main receptors [33].

Semaphorins were originally identified as axon guidance molecules required for axon guidance [7]. Studies have shown that they also play vital roles in diverse physiological and pathological processes, including cardiomyogenesis [34,35], tumor neovascularization and metastasis [19,36,37,38], bone remolding [39,40,41,42], angiogenesis [15,42,43,44,45], and immunomodulation [46,47,48]. The representative recent studies on the role of semaphorins in diseases are listed in Table 1. In recent years, several semaphorin members have been reported to participate in metabolic disorders. In this review, we summarize the progress on the role of semaphorins in metabolic disorders including obesity, adipose inflammation, brown adipose tissue, immune cell metabolism, as well as diabetic complications.

## 2. Semaphorins in Metabolism

Metabolic disorders are becoming a worldwide pandemic, with increased incidence of obesity and related diseases. Semaphorins have been reported to participate in many aspects of metabolism, from obesity to adipose inflammation. Semphorins also regulate the function of brown adipose tissue and immune cell metabolism (summarized in Table 2).

### 2.1. Semaphorins in Obesity

Obesity is becoming a worldwide health threat. Obesity is an important risk factor for cardiovascular [78] and metabolic diseases, including type 2 diabetes [79], and can also affect women’s fertility and human reproduction [80]. Several studies have recently identified semaphorins as an important regulator of obesity by regulating adipogenesis or hypothalamic melanocortin circuit development.

#### 2.1.1. Semaphorins in Adipogenesis

Two semaphorin members have been reported to regulate adipogenesis. In 2016, Liu et al. [69] reported that Sema3A promoted adipose mesenchymal stem cells (ASCs) towards osteogenic phenotype and played an inhibitory role in adipogenesis of ASCs. Sema3A also decreased the expression of adipose-related genes FABP4, PPARγ, and CEBPα, as well as lipid droplet formation (Figure 2).

Sema3G is a secreted semaphorin highly expressed in adipocytes and can be induced by PPARγ in endothelial cells [81]. Sema3G utilizes Nrp1, but not Nrp2, as a receptor to induce the repulsion of sympathetic axons [82]. Earlier this year, Liu et al. [70] showed that Sema3G was upregulated during adipogenesis, and overexpression of Sema3G significantly enhanced adipogenesis of 3T3-L1 cells and primary preadipocytes. The promoting effect of Sema3G in adipogenesis is partly prevented by anti-Nrp2 antibodies, suggesting that Sema3G promotes adipocyte differentiation through Nrp2 receptor. On the contrary, Sema3G knockdown by specific small hairpin RNA (shRNA) in 3T3-L1 cells and primary mouse preadipocytes decreased adipocyte differentiation. Sema3G knockdown reduced weight gain, fat mass, and lipogenesis in liver, and ameliorated insulin resistance and glucose tolerance in mice on HFD. Additionally, they proved that Sema3G inhibited HFD-induced obesity through the PI3K/Akt/GSK3β signaling pathway in adipose tissue and the AMPK/SREBP-1c pathway in the liver. Clinical investigation have shown an increase in Sema3G levels in the plasma of obese people as compared with non-obese individuals (Figure 2) [70].

#### 2.1.2. Semaphorins in Hypothalamic Regulation of Obesity

Dysregulation in the central nervous system could lead to obesity [83]. The hypothalamus nucleus controls food intake and energy expenditure. There are two neuronal populations in the arcuate nucleus of the hypothalamus (ARH), i.e., the proopiomelanocortin (POMC) neurons suppressing appetite, and the neuropeptide Y (NPY)/agouti-related peptide (AgRP) neurons inducing appetite [84]. The melanocortin receptor 4 (MC4R) locus has been shown to have a strong association with body mass index (BMI) in genome-wide association studies (GWAS). MC4R regulates feeding behavior in the paraventricular nucleus of the hypothalamus (PVH) [85]. The POMC and AgRP neurons in the ARH can project into the PVH and have opposite effects on the MC4R expressing neurons [86]. Nutritional status including undernutrition, overnutrition, and maternal high-fat diet feeding, affects ARH-PVH circuits and diet-induced obesity [87].

In a recent paper, van der Klaauw et al. examined the role of Class-3 semaphorins signaling in the development of ARH-PVH circuits. They found 40 rare variants in Sema3A–3G and their receptors (Plexin1–4 and Nrp1 and -2) in severely obese individuals. Rare variants in these genes were enriched in 982 severely obese individuals as compared with the controls. Deletion of seven Class-3 semaphorin members in zebrafish has led to adiposity. Sema3 members expressed in the hypothalamic PVH, and their receptors Nrps and plexins expressed on the POMC^+^ neurons in the ARH. In mice, neuropilin-2 receptor (Nrp2) deletion in POMC neurons disrupted their projections from the ARH to the PVH, reduced energy expenditure, and caused weight gain. Cumulatively, the results showed that Class-3 semaphorins contributed to the development of hypothalamic melanocortin circuits involved in energy homeostasis and obesity (Figure 3) [71,88].

### 2.2. Semaphorins in Adipose Inflammation

Obesity is usually accompanied with chronic low-grade inflammation. The expanding adipose undergoes extensive remodeling, leading to infiltration of macrophages that secrete proinflammatory cytokines and contribute to the development of systemic insulin resistance [89].

A study by Shimizu et al. identified Sema3E as a regulator of adipose tissue macrophage accumulation in obesity that contributes to systemic insulin resistance [90]. Sema3E and its receptor PlexinD1 were upregulated in the adipose tissue of a dietary obesity mouse model. Expression of Sema3E was induced by p53 in adipocytes and promoted the influx of monocyte-derived macrophages into the visceral white adipose tissue through its receptor PlexinD1. Inhibition of the Sema3E-PlexinD1 axis markedly reduced adipose tissue inflammation and improved insulin resistance, whereas overexpression of Sema3E in adipose tissue promoted infiltration of macrophages, adipose inflammation, and insulin resistance (Figure 4) [72]. In 2019, the same lab developed a peptide vaccine for Sema3E. Two peptides (HKEGPEYHWS) conjugated to keyhole limpet hemocyanin (KLH) were injected into mice to generate neutralizing antibodies for Sema3E. The Sema3E antibody titer increased after injection of the KLH-conjugated Sema3E peptide, and suppressed the infiltration of PlexinD1 positive cells, ameliorated visceral adipose tissue inflammation, and systemic glucose intolerance, suggesting that Sema3E peptide vaccine has therapeutic potential for obesity and diabetes [73].

Mejhert et al. identified Sema3C as a new adipokine regulated by weight changes and its expression correlated significantly with body weight, insulin resistance, and adipose tissue morphology (hypertrophy vs. hyperplasia). In preadipocytes, Sema3C enhanced the production and secretion of several extracellular matrix components (fibronectin, elastin, and collagen I) and matricellular factors (connective tissue growth factor, IL6, and transforming growth factor-β1). Furthermore, the expression of Sema3C in human white adipose tissue (WAT) correlated positively with the degree of fibrosis in WAT. These results suggest that Sema3C constitutes an adipocyte-derived paracrine signal that influences ECM composition and could play a pathophysiological role in human WAT fibrosis [74].

### 2.3. Semaphorins in Brown Adipose Tissue

There are two types of adipose tissue, WAT and brown adipose tissue (BAT). WAT includes visceral adipose tissue (v-WAT) and subcutaneous adipose tissue (s-WAT). BAT and WAT originate from different progenitor cells [91]. BAT cells are filled with a high abundance of mitochondria that can oxidize fatty acid and generate heat [92]. Semaphorins are expressed and functional in brown adipose tissue. In 2001, Antonio Giordano et al. found that both active isoforms of Sema3A were expressed in rat interscapular BAT and that cold-acclimation inhibited the secretion of Sema3A in the brown adipocytes of rats [75]. Mass spectrometry (MS)-based phosphoproteomic screening of brown preadipocytes in the basal and IGF-1 stimulated states verified the induction of phosphorylation on Sema4B, suggesting its possible role of insulin/IGF-1 signaling in brown adipocytes [77].

Another study showed that the plexin/semaphorin axis played a role in macrophage–axon crosstalk in BAT. Macrophages in BAT provided immunological defense and contributed to the control of tissue innervation. Disruption of this circuit in BAT resulted in metabolic imbalance. Mice with specific Mecp2 deletion in macrophages led to spontaneous obesity and impaired BAT function by disrupting sympathetic innervation and local release of norepinephrine. Mecp2-deficient macrophages showed significant upregulation of PlexinA4 as compared with Mecp2-sufficient macrophages. Sema6A was expressed by sympathetic nerves in BAT. PlexinA4 inhibited the axonal outgrowth of Sema6A-positive nerves and innervation of BAT [76].

### 2.4. Semaphorins in Immune Cell Metabolism

Polarization of macrophages towards proinflammatory or anti-inflammatory states has different metabolic requirements [93]. Sema6D deletion inhibited anti-inflammatory polarization of macrophages, accompanied by decreased PPARγ expression, fatty acid uptake, and lipid metabolic reprogramming. Sema6D mediated anti-inflammatory polarization through macrophage PlexinA4. The cytoplasmic region of Sema6D associated with C-Abl, a tyrosine kinase, to stimulate PPARγ expression. These findings demonstrate the role of Sema6D signaling in macrophage polarization, coupling immunity, and metabolism via PPARγ [59].

## 3. Semaphorins in Diabetic Complications

Type 2 diabetes is a metabolic disorder syndrome that occurs over a prolonged period of time and can cause many other diseases [94]. Multiple complications are associated with diabetes and most patients with diabetes die of diabetic complications. Prevention and treatment of diabetic complications is of great clinical significance [95]. Many semaphorins are involved in diabetic complications. The functions of semaphorins in diabetic complications are summarized in Table 3.

### 3.1. Semaphorins in Diabetic Retinopathy

Diabetic retinopathy (DR) is a common complication of diabetes and leads to blindness [119]. Studies have indicated that several semaphorins are upregulated in DR, contributing to neurovascular pathophysiology of DR, and remain as an intense investigation topic for DR (Figure 5) [120].

Sema3A [99] is induced in ischemic retinal ganglion cells in response to IL-1β after vascular injury, and prevents revascularization of ischemic but salvageable neurons [100]. Sema3A binds Nrp1 to mediate endothelial cell cytoskeleton collapse and prevent migration [121], and promotes endothelial apoptosis [122]. Joyal et al. demonstrated that inhibition of Sema3A facilitated normal revascularization of the inner retina after vascular injury in oxygen-induced retinopathy (OIR) mice [100]. Sema3A levels have been significantly elevated in the vitreous of patients with diabetic macular edema [98] and proliferative DR [99]. Serum Sema3A levels have also correlated with the phenotypes of diabetic retinopathy [97]. In early DR, Sema3A induces vascular hyperpermeability through Nrp1, and then activates Src kinase and FAK, loosens endothelial tight junctions, and facilitates blood-retinal barrier breakdown (Figure 5) [98]. Sema3A neutralization alleviates vascular hyperpermeability in early DR, at a stage when anti-VEGF therapy is ineffective [98]. In addition, neuron-derived Sema3A provokes microglial chemotaxis through Nrp1 and can contribute to inflammation associated with retinopathy [99]. Sema3A can also promote apoptosis in retinal neurons, contributing to neurodegeneration in retinopathy. Hua et al. demonstrated that a Sema3A-neutralizing antibody alleviates neurodegeneration in OIR [123].

Sema3E is secreted by severely ischemic retinal neurons and activates the endothelial cell PlexinD1 to deter VEGF-induced filipodial projections in an OIR model [124]. However, Sema3E normalizes revascularization in OIR and suppresses extra retinal vascular outgrowth without affecting normal regeneration of the retinal vasculature (Figure 5) [101]. Sema3E expression also decreased in the vitreous of DR patients [101].

Jie-hong Wu et al. [96] found increased expression of Sema4D mRNA on the retinas of OIR and DR mouse models. Soluble Sema4D levels have increased significantly in the aqueous fluid of DR patients and were negatively correlated with the success of anti-VEGF therapy. The upregulation of Sema4D is regulated by IRF1. Sema4D can be cleaved by ADAM17 to generate soluble isoforms [125]. Sema4D promotes endothelial cell migration and permeability through the PlexinB1/mDIA1/Src signaling. The signaling pathway also induces pericyte migration and N-cadherin internalization to worsen vascular permeability (Figure 5). Inhibition of the Sema4D/PlexinB1 pathway by genetic deletion or anti-Sema4D antibody reduces pericyte loss and vascular leakage in a STZ induced DR model and inhibits neovascularization in an OIR model. Furthermore, the anti-Sema4D antibody has synergistic therapeutic effect with the anti-VEGF antibody. This indicates that the therapeutic effect of Sema4D antibody can be used to complement or improve the treatment of DR [96].

### 3.2. Semaphorins in Diabetic Nephropathy

The onset of diabetic nephropathy (DN) is highlighted by glomerular filtration barrier abnormalities [126]. Podocytes are complex epithelial cells that help keep the integrity and function of the kidney glomerular filters [127]. Several semaphorins are expressed in podocytes and regulate their function in DN.

Sema3A is secreted by podocytes and excess Sema3A disrupts the glomerular filtration barrier. Pardeep et al. showed that Sema3A was upregulated in the podocyte of DN patients. Sema3A signaling has been shown to regulate podocyte shape, induce glomerular disease, and aggravate DN through nephrin, αvβ3 integrin, and MICAL1 interactions with plexinA1 [104]. Urinary Sema3A also increased as early as two weeks after induction of diabetes and increased over time in conjunction with the development of nephropathy. Consistent with the data from animal studies, increased Sema3A urinary excretion has been detected in diabetic patients with albuminuria, particularly in those with macroalbuminuria. Genetic ablation of Sema3A, or pharmacological inhibition with a novel Sema3A inhibitory peptide, protected against diabetes-induced albuminuria, kidney fibrosis, inflammation, oxidative stress and renal dysfunction. Sema3A could be a biomarker and a mediator of DN [102]. Podocyte Vegf^164^ overexpression has increased VEGF receptor 2 and Sema3A levels and dramatically worsened diabetic nephropathy in a streptozotocin induced mouse model of diabetes [103].

Sema3G is also expressed in podocytes [106]. Ultrastructural analyses have revealed partially aberrant podocyte foot process structures, but not obvious glomerular defects in Sema3G deficient mice. When challenged with lipopolysaccharide to induce acute inflammation, or streptozotocin to induce diabetes, Sema3G deletion has resulted in increased albuminuria. On the one hand, podocyte specific Sema3G deletion has enhanced chemokine ligand 2 and interleukin 6 expression. On the other hand, Sema3G overexpression has attenuated inflammatory cytokine expression through the inhibition of lipopolysaccharidinduced Toll-like receptor 4 signaling. Therefore, Sema3G can protect podocytes from inflammatory kidney diseases and diabetic nephropathy [106].

Further studies have shown dysregulation of Sema5A and Sema5E in diabetic nephropathy [105]. A large transcriptional dataset on human diabetic glomeruli shows Sema5A and Sema3G are among the top 100 dysregulated transcripts and “semaphorin signaling in neurons” as one of the enriched pathways [128]. Francesco Sambo et al. analyzed novel genetic susceptibility loci for diabetic end-stage renal disease identified through robust naive Bayes classification. Sema6D was associated with end-stage renal disease (ESRD) in the FinnDiane study [107].

### 3.3. Semaphorins in Diabetic Neuropathy

Diabetic peripheral neuropathy (DPN) is a common complication with altered sensation as a result of damage to peripheral sensory nerves, which is preferentially affected in the early stages of diabetes, and is the primary cause of diabetes-related hospital admissions and nontraumatic foot amputations [129].

All Class-3 semaphorins are expressed in the cornea and Sema3A expression increased fast upon cornea injury. In isolated adult trigeminal ganglia or dorsal root ganglia neurons, Sema3A produced similar neuronal growth in cells treated with neural growth factor (NGF), and the length of the neurites and branching were comparable between both treatments. Mice receiving intrastromal pellet implantation containing Sema3A showed an enhanced corneal nerve regeneration as compared with those receiving pellets with vehicle. In adult peripheral neurons, Sema3A is a potent inducer of neuronal growth in vitro and cornea nerve regeneration in vivo [111].

Sema3A is also produced by keratinocytes and has a chemorepulsive effect on intraepidermal nerve fibers. High glucose upregulates Sema3A in diabetic keratinocytes via the mTOR-mediated p70-S6K and 4EBP1 signaling pathways. Higher Sema3A expression and overactivation of mTOR signaling were accompanied with reduced intraepidermal nerve fiber density (IENFD) in the skin of diabetic patients as compared with control subjects. This pathway could play a critical role in diabetic small fiber neuropathy (SFN) [110].

The diabetic cornea exhibits pathological alterations, such as delayed epithelial wound healing and nerve regeneration. Wounding induced the expression of Sema3A, Sema3C, and their receptor Nrp2 in normal corneal epithelial cells, but the upregulation was inhibited in diabetic cornea. Exogenous Sema3C resulted in an increased rate of wound healing and nerve fiber regeneration, while Sema3C shRNAs and Nrp2-neutralizing antibodies had opposing effects to Sema3C in diabetic cornea [109]. Sema3C is also highly expressed in inflamed tissue in Charcot foot patients [108].

Exosomes derived from healthy Schwann cells (SC-Exos) show therapeutic effects for type 2 diabetic peripheral neuropathy by improving sciatic nerve conduction velocity and increasing thermal and mechanical sensitivity. Western blot analysis of sciatic nerve tissues showed that the DPN considerably increased Sema6A expression, whereas the SC-Exos treatment, significantly reduced Sema6A expression, suggesting that Sema6A could contribute to DPN [112].

### 3.4. Semaphorins in Diabetic Wound Healing

Chronic wound is also a common and severe long-term diabetic complication. The diabetic foot is the most common complication and the main cause of amputations associated with nonhealing ulcers [130]. Angiogenic regulators are important for tissue repair in diabetes mellitus. Wang et al. investigated the effects of soluble Sema4D on wound healing in db/db diabetic mice. Their results showed that Sema4D accelerated wound healing in diabetic mice by promoting angiogenesis and reducing the inflammatory response [113]. Sema6A suppression by microRNA miR-27b rescued impaired bone marrow-derived angiogenic cell (BMAC) angiogenesis and accelerated wound healing in type 2 diabetic mice [114].

### 3.5. Diabetic Osteoporosis

Diabetes mellitus causes diabetic osteoporosis, a chronic bone metabolic disease, which is characterized by an increased risk of osteoporotic fracture and deterioration of bone microarchitecture [131]. Diabetic rats exhibited a pronounced bone phenotype which manifested by decreased expression of Sema3A, IGF-1, and β-catenin, as well as PPARγ, suggesting that the Sema3A-IGF1-β-catenin pathway was involved in the alterations of bone microarchitecture and bone strength of diabetic rats. Sema3A deficiency in bone can contribute to upregulation of PPARγ and cathepsin K, which further disrupts bone remodeling in diabetic rats [115]. Qiao et al. also found decreased expression of Sema3A in bone mesenchymal stem cells (BMSC) derived from diabetes rats. Stimulating with Sema3A significantly increased the expression of osteogenic related genes, including type I collagen, alkaline phosphatase, runt related transcription factor 2 (RUNX2), bone morphogenetic protein, and osteocalcin. Additionally, the osteogenic capacity of BMSCs was also increased by Sema3A stimulation [116]. Sema3A pretreated BMSC [117] or adipose mesenchymal stem cell (ASC) [118] sheets show therapeutic effects for new bone formation in type 2 diabetes mellitus rats.

## 4. Miscellaneous

Semaphorins can regulate the metabolism of amino acids. Toshinori Sawano et al. [132] reported that Sema4D regulated microglial proliferation at least in part by regulating the competitive balance of L-arginine metabolism. In activated microglia, L-arginine is metabolized competitively by inducible nitric oxide synthase (iNOS) and arginase (Arg). iNOS synthesizes NO and Arg turns L-arginine into plyamines. Sema4D deficiency altered the balance of L-arginine metabolism between iNOS and Arg, leading to an increase in the production of polyamines in middle ceral artery occlusion, while its presence inhibited polyamine production in primary microglia obtained from Sema4D-/-mice. Sema4D regulates the metabolism of L-arginine by inhibiting the activity of Arg1.

The islets of Langerhans are endocrine organs that secrete insulin and become malfunctioned in metabolic disorders. The Sema3A-Nrp2 axis constitutes a chemoattractant system essential for the development of pancreatic islets. In the fetal pancreas, peripheral mesenchymal cells express Sema3A, whereas central nascent islet cells produce its receptor Nrp2. Nrp2 mutant islet cells have defects in migration and are unresponsive to purified Sema3A. Mutant Nrp2 islets aggregate centrally and fail to disperse radially [133]. Sema4C and Sema3E have been found to change in sialylation in glucose stimulated islet cells [134]. PlexinB2, which could act as a receptor for Sema4D and Sema4C localizes to the pancreatic islets of Langerhans [135]. There are also studies on the role of semaphorins in pancreatic endocrine tumors of islets [136]. These studies suggest a possible role of semaphorins in islet function and diabetes, which could be uncovered in future studies.

## 5. Perspectives

With the change towards a westernized lifestyle, the incidence of obesity and metabolic disorders is increasing worldwide, which ranges from obesity to type 2 diabetes, leading to complications in the kidney, retina, and foot. In this review, we summarized the role of semaphorins and their underlying mechanisms in metabolic disorders including obesity, adipose inflammation, diabetic complications. Since semaphorins express in a variety of cells, they participate in many aspects of metabolic diseases. Several Sema3 members have been reported to regulate obesity by regulating adipogenesis (with an inhibitory role for Sema3A and a stimulatory role for Sema3G) and hypothalamic melanocortin circuits development (mutations of Sema3 and their receptors resulting in early onset of obesity). Semaphorins also contribute to adipose tissue inflammation, which is a main cause of insulin resistance. In diabetic complications, semaphorins could have stimulatory or inhibitory roles in the development of diabetic retinopathy, nephropathy, neuropathy, wound healing and osteoporosis by regulating immune responses, peripheral neural growth, angiogenesis, and other mechanisms.

Although investigations on the role of semaphorins in metabolic diseases are in the infantile state, evidence from translational studies has been emerging. Products targeting semaphorins have been used for prevention and therapy of adipose inflammation and diabetic complications. In 2019, Yohko Yoshida et al. developed a Sema3E peptide vaccine that led to the generation of neutralizing antibodies for Sema3E, and thus suppressed visceral adipose tissue inflammation and systemic glucose intolerance, suggesting that Sema3E peptide vaccine has therapeutic potential for obesity and diabetes [73]. Neutralization antibodies of Sema3A alleviates vascular hyperpermeability in early diabetic retinopathy [98] and neurodegeneration in OIR [123]. The anti-Sema4D antibody has a synergistic therapeutic effect with the anti-VEGF antibody to improve the treatment of diabetic retinopathy [96]. Pharmacological inhibition with a novel Sema3A inhibitory peptide has been shown to protect against diabetic nephropathy [102]. Sema3A pretreated BMSC [117] or adipose mesenchymal stem cells (ASC) [118] have shown therapeutic effects for new bone formation in type 2 diabetes mellitus rats. Future studies on the role of additional semaphorin family members in metabolic disorders and their potential network regulation would promote a full understanding regarding the functional role of semaphorins in metabolic diseases, which could provide potential new therapeutic targets and pharmacological intervening approaches.

## Figures and Tables

**Figure 1 ijms-21-05641-f001:**
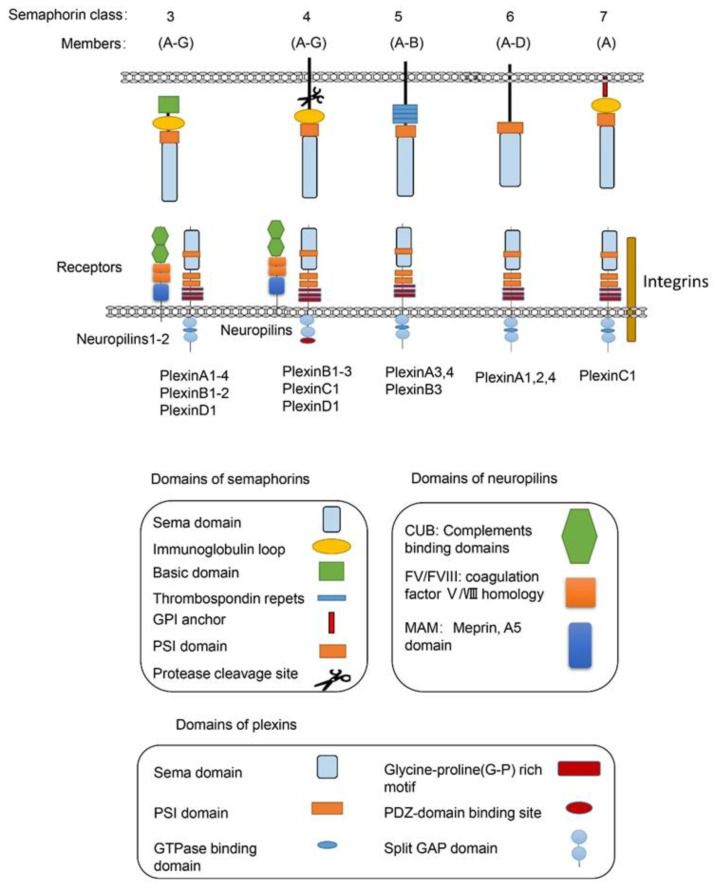
The vertebrate semaphorins and their main receptors. The main structural features of the subfamilies of the vertebrate semaphorins and their receptors are shown. The members of the vertebrate semaphorin family all contain the hallmark Sema domain and are divided into five subfamilies based upon structural features. The Class-3 semaphorins are the only secreted semaphorins. The main receptors for semaphorins are listed below. For most Class-3 semaphorins, neurophilins and plexins are their receptors. For Class-4 semaphorins, the main receptors are PlexinB molecules, while PlexinC1, PlexinD1, and neuropilins can also act as their receptors. For the only member of Sema7, PlexinC1 and integrin β1 are the main receptors. Many other molecules including integrin, proteoglycans, and RTKs have also been included as receptors for semaphorins, which are not showed in the diagram.

**Figure 2 ijms-21-05641-f002:**
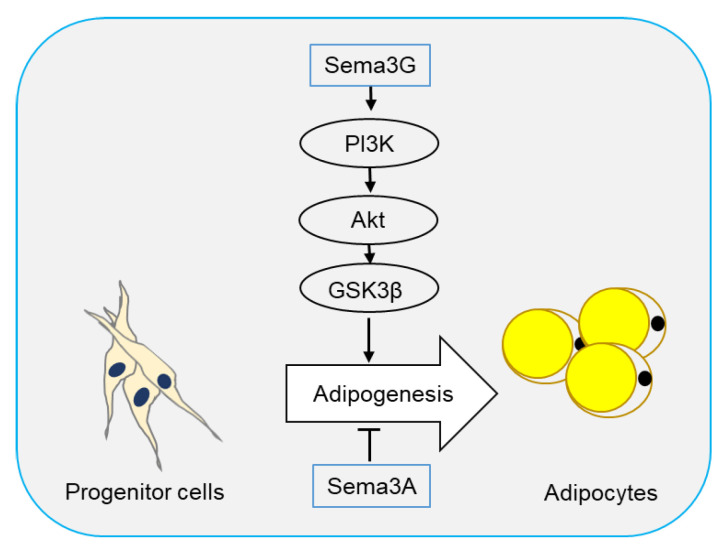
Semaphorins in adipogenesis. Adipocytes originate from progenitor cells through a process called adipogenesis. Sema3G has been reported to stimulate adipogenesis through the PI3K/Akt/GSK3β signaling pathway, while Sema3A plays an inhibitory role.

**Figure 3 ijms-21-05641-f003:**
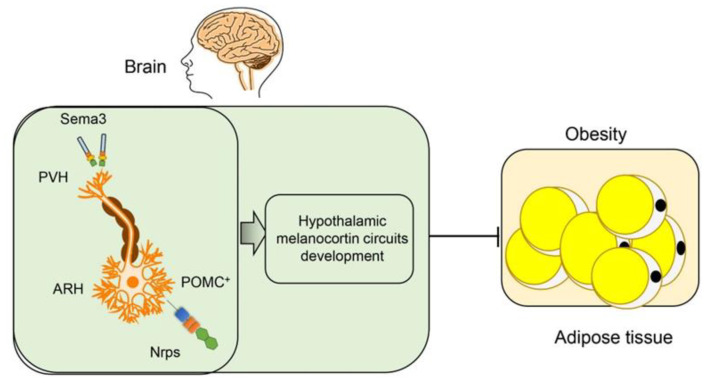
Semaphorins in the hypothalamic regulation of obesity. Sema3 family members are expressed in the hypothalamic paraventricular nucleus of the hypothalamus (PVH), and their receptors neuropilins (Nrps) and plexins are expressed on proopiomelanocortin (POMC+) neurons in the arcuate nucleus of the hypothalamus (ARH). The Sema3-Nrp axis helps the projections of POMC neurons toward the PVH and the development of hypothalamic melanocortin circuits, which inhibits obesity.

**Figure 4 ijms-21-05641-f004:**
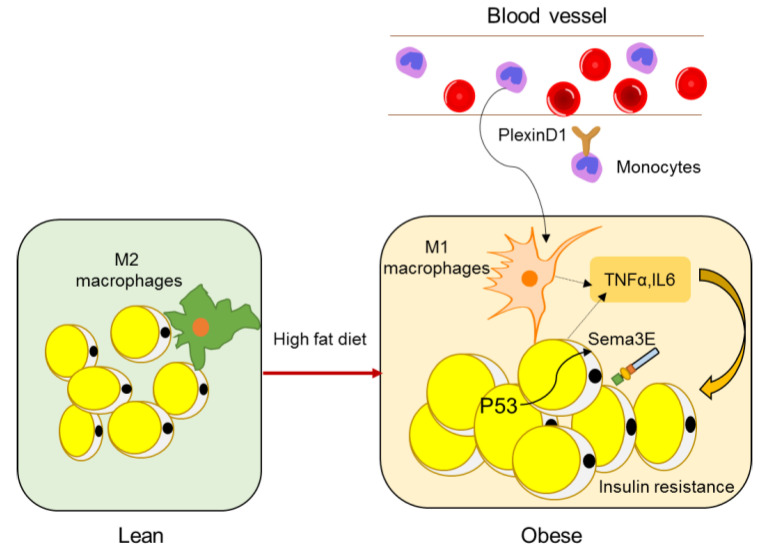
Semaphorins in adipose inflammation. In lean adipose tissue, the main macrophages are quiescent M2 macrophages, whereas, in obese adipose tissue, monocytes from peripheral blood are recruited to adipose tissue and macrophages become M1 polarized macrophages (other immune cell types such as T cells are not shown here), leading to an inflammatory state, which is a main cause of insulin resistance. Expression of Sema3E is induced by p53 in adipocytes and promotes the influx of monocyte-derived macrophages through its receptor PlexinD1.

**Figure 5 ijms-21-05641-f005:**
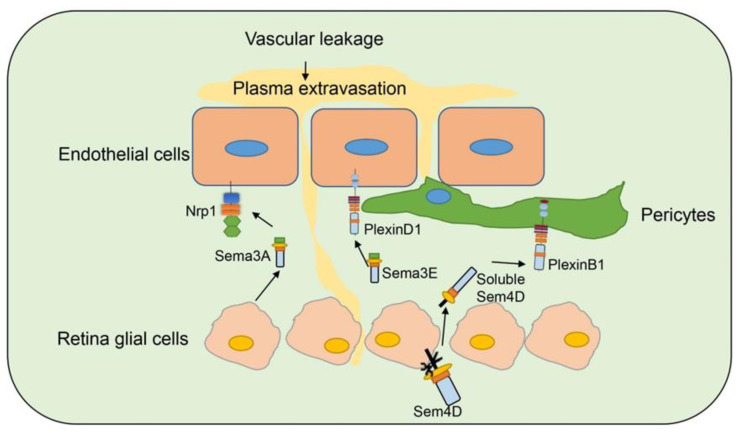
Semaphorins in diabetic retinopathy. Several semaphorin family members are involved in diabetic retinopathy. Sema3A is secreted by retina glia cells and bind to Nrp1 on endothelial cells to induce vascular permeability. Sema3E is secreted by neural cells in retina and binds to PlexinD1 on endothelial cells and normalizes angiogenic directionality in the retina. Sema4D on retina glia cells can be cleaved and secreted as soluble Sema4D. Soluble Sema4D binds to PlexinB1 on pericytes and induces pericyte loss and vascular leakage.

**Table 1 ijms-21-05641-t001:** Representative recent studies on the role of semaphorins in diseases *.

Diseases	References	Major Findings	Semaphorins Involved
Cancer	Lee, Munuganti et al., 2018 [49]	Promising small molecule inhibitors bind to SEMA3C, and attenuate prostate cancer growth	Sema3C
	Jiang, Chen et al., 2016 [50]	Sema4D influences cell proliferation, invasion, migration, and apoptosis of breast cancer cells	Sema4D
	Tarullo, Hill et al., 2020 [51]	Sema7A promotes breast cancer progression	Sema7A
Angiogenesis	Lee, Kim et al., 2018 [52]	Therapeutic Sema3A antibody F11 attenuated angiogenesis in glioblastoma	Sema3A
	Yang, Zeng et al., 2019 [53]	Sema4C promotes angiogenesis in breast Cancer	Sema4C
	Chen, Zhang et al., 2018 [54]	Sema4D has synergistic effects with vegf on the promotion of angiogenesis	Sema4D
Multiple sclerosis	Gutierrez-Franco, Eixarch et al., 2017 [55]	Sema7A is involved in peripheral immunity and CNS inflammation in MS pathogenesis	Sema7A
Rheumatoid arthritis	Xie and Wang 2017 [56]	Sema7A promotes rheumatoid arthritis	Sema7A
	Yoshida, Ogata et al., 2015 [57]	Sema4D Contributes to rheumatoid arthritis by inducing inflammatory cytokine production	Sema4D
Colitis	Eissa, Hussein et al., 2019 [58]	Sema3E regulates apoptosis in the intestinal epithelium and inhibits colitis	Sema3E
	Kang, Nakanishi et al., 2018 [59]	Sema6D is important for generation of intestinal resident CX3CR1^hi^ macrophages and prevents development of colitis	Sema6D
	Delgoffe, Woo et al., 2013 [60]	Sema4A regulates established inflammatory colitis through Nrp1	Sema4A
Bone remodeling	Kenan, Onur et al., 2019 [61]	Sema3A prevents bone resorption by inhibiting osteoclasts and increases bone formation by inducing osteoblasts	Sema3A
Endocrine diseases	Parkash, Messina et al., 2015 [62]	Sema7A promotes projection of gonadotropin-releasing hormone (GnRH) neurons and maintains normal oestrous cyclicity and fertility	Sema7A
	Oleari, Caramello et al., 2019 [63]	Sema3A regulates Gonadotropin-releasing hormone neurons through its receptor neuropilins, PlexinA1 and PlexinA3	Sema3A
Atherosclerosis	Hu, Liu et al., 2018 [32]	Disturbed flow regulated Sema7A promotes atherosclerosis	Sema7A
	Wu, Li et al., 2017 [64]	Sema3E attenuates neoinitimal formation via suppressing VSMCs migration and proliferation	Sema3E
Cardiovascular diseases	Sun, Peng et al., 2019 [65]	Sema6D regulates perinatal cardiomyocyte proliferation and maturation in mice	Sema6D
	Sandireddy, Cibi et al., 2019 [66]	Sema3E-PlexinD1 signaling is required for cardiac ventricular compaction	Sema3E
Neuronal diseases	Lee, Macpherson et al., 2017 [67]	Sema3A and Sema7A regulates bitter and sweet neurons, respectively	Sema7A Sema3A
	Frias, Liang et al., 2019 [68]	Sema4D induces inhibitory synapse formation	Sema4D

***** Not all the semaphorin molecules involved in the diseases are listed here.

**Table 2 ijms-21-05641-t002:** Semaphorins in metabolism.

Metabolic Functions	Semaphorins Involved	Receptors Involved	References
Adipogenesis	Sema3A	Not mentioned	[69]
	Sema3G	Nrp 2	[70]
Hypothalamus regulation of obesity	Sema3A, Sema3B, Sema3C, Sema3D Sema3E, Sema3F, Sema3G	PlexinA1, PlexinA2, PlexinA3, PlexinA4, Nrp1, Nrp2	[71]
Adipose inflammation and fibrosis	Sema3E	PlexinD1	[72,73]
	Sema3C	Not mentioned	[74]
Brown adipose tissue function	Sema3A	Not mentioned	[75]
	Sema6A	PlexinA4	[76]
	Sema4B	Not mentioned	[77]
Immune cell metabolism	Sema6D	PlexinA4	[59]

**Table 3 ijms-21-05641-t003:** Semaphorins in diabetic complications.

Diabetic Complications	Semaphorins Involved	Receptors Involved	References
Diabetic retinopathy	Sema4D	PlexinB1	[96]
	Sema3A	Nrp1	[97,98,99,100]
	Sema3E	PlexinD1	[101]
Diabetic nephropathy	Sema3A	Nrp1, PlexinA1	[97,102,103,104]
	Sema3E	Not mentioned	[105]
	Sema3G	Not mentioned	[105,106]
	Sema5A, Sema5E	Not mentioned	[105]
	Sema6D	Not mentioned	[107]
Diabetic neuropathy	Sema3C	Nrp1, Nrp2	[108,109]
	Sema3A	Not mentioned	[110,111]
	Sema6A	Not mentioned	[112]
Diabetic wound healing	Sema4D	PlexinB2	[113]
	Sema6A	Not mentioned	[114]
Diabetic osteoporosis	Sema3A	Not mentioned	[115,116,117,118]

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
