# Peer review of "The Role of Semaphorins in Metabolic Disorders"

_ijms, 2020, doi:10.3390/ijms21165641_

Round 1

Reviewer 1 Report

The present review entitled “Role of semaphorins in metabolic disorders” is an interesting work for researchers working in metabolic and lifestyle factors research. Authors highlighted the role of semaphorins in metabolic disorders with an appropriate historical background. Authors have presented this review quite well and the message of the text is quite clear. A review in the area would be of great interest to the common readership of the journal.

Following suggestions could improve the manuscript:

  1. Add reference for line 26-27.
  2. I suggest authors may include a table summarizing the most recent studies reported on semaphorins for several diseases (Experimental studies).
  3. The presentation of the Manuscript is very well. I believe some possible inclusion of bioinformatics approaches of semaphorins (if possible, to do) and signalling pathways will improve the paper quality.
  4. Authors should provide detail paragraph on the applications and future perspectives of semaphorins in metabolic disorders.
  5. English and grammar level in the manuscript is good enough for publication.

Author Response

1.     Add reference for line 26-27.

Response: Thanks. References were added.

2.     I suggest authors may include a table summarizing the most recent studies reported on semaphorins for several diseases (Experimental studies).

Response: We agree with the reviewer’s point. We created a table about the role of semphorins in several diseases as the new table 1 in page 3.

3.     The presentation of the Manuscript is very well. I believe some possible inclusion of bioinformatics approaches of semaphorins (if possible, to do) and signalling pathways will improve the paper quality.

Response: We agree with the reviewer’s point. Indeed, bioinformatics approaches are very helpful for the understanding of the role of semaphorins in metabolic disorders. We may conduct bioinformatics investigation on the role of semaphorins in metabolic diseases in the future studies. Here we add some more detailed signaling pathway information in the manuscript as the reviewer suggested (shown in Figure 2 and Figure 4).

4.     Authors should provide detail paragraph on the applications and future perspectives of semaphorins in metabolic disorders.

Response: Thanks for your good suggestion. We revised our Perspectives paragraph at the end of the review as the reviewer suggested (mostly from line 371 to line 404).

5.     English and grammar level in the manuscript is good enough for publication

Response: Thanks.

Reviewer 2 Report

The authors have published an interesting review of the Role of semaphorins in metabolic disorders.
The authors suggest a possible role of semaphorins in islet and diabetes, which will be uncovered in future studies.
I would like to receive additional information about the relevance and novelty study of semaphorins in the diabetes
The authors should correct mistakes in the text.

Author Response

The authors have published an interesting review of the Role of semaphorins in metabolic disorders.

The authors suggest a possible role of semaphorins in islet and diabetes, which will be uncovered in future studies.

I would like to receive additional information about the relevance and novelty study of semaphorins in the diabetes

The authors should correct mistakes in the text.

Response:

Thank you for your great suggestion. We are very interested in the role of semaphorins in diabetes. Several semaphorin members or their receptors are expressed in islet cells and Sema3A-Nrp2 axis regulates the development of pancreatic islets. However, there has been limited literature about the role of semaphorins in the functions of islets so far. we believe more researchers will be interested in this field in the future. Type 2 diabetes, which is a disease caused by multiple factors, usually starts from systemic metabolic disorder. Semaphorins participate in the pathogenesis of obesity, adipose inflammation and insulin sensitivity, which we have discussed in this review. Most researchers are more concerned about the complications of diabetes, which are affected in several ways by semaphorin family members. In regard to the mistakes in the text, we have tried our best to correct as much as we could.
